# Colour Response in Western Flower Thrips Varies Intraspecifically

**DOI:** 10.3390/insects13060538

**Published:** 2022-06-10

**Authors:** Karla Lopez-Reyes, Karen F. Armstrong, David A. J. Teulon, Ruth C. Butler, Coby van Dooremalen, Monika Roher, Robert W. H. M. van Tol

**Affiliations:** 1Bio-Protection Research Centre, Lincoln University, Lincoln 7647, New Zealand; karen.armstrong@lincoln.ac.nz; 2New Zealand Institute for Plant and Food Research Limited, Private Bag 4704, Christchurch 8140, New Zealand; david.teulon@plantandfood.co.nz (D.A.J.T.); rcb962@gmail.com (R.C.B.); 3Bees@wur, Wageningen University and Research Centre, Droevendaalsesteeg 1, 6708 PB Wageningen, The Netherlands; coby.vandooremalen@wur.nl (C.v.D.); rob@bugresearch.nl (R.W.H.M.v.T.); 4Faculty of Agriculture, University of Zagreb, Svetošimunska cesta 25, 10000 Zagreb, Croatia; mroher95@gmail.com; 5BugResearch Consultancy, Ambachtstraat 29, 6707 JJ Wageningen, The Netherlands

**Keywords:** *Frankliniella occidentalis*, colour choice-test, behaviour, trapping, wind tunnel, thrips

## Abstract

**Simple Summary:**

The western flower thrips (WFT) is a major glasshouse pest worldwide. Colour and odour are used to attract WFT to traps for monitoring in pest management. However, the use of colour attraction for surveillance, or its potential as a control tool for mass trapping, has not progressed due to ongoing debate about the optimal colour. Blue is commonly used for monitoring, but yellow can also be preferred over blue, confounding the implications for field application. As methodological differences amongst studies could cause this discrepancy, in the present study we evaluated populations from Germany (DE) and the Netherlands (NL) that had previously expressed different preferences for blue and yellow. Examining the behavioural response of WFT to blue or yellow LEDs in a wind tunnel by using the same methodology and facilities, the two populations maintained their original differential preferences. This remained after several generations under common conditions, although a significant increase in attraction to yellow was observed for both populations when reared on yellow-flowered chrysanthemum plants. This is the first report demonstrating that WFT populations can differ in their colour preferences even under the same experimental circumstances. Given that research is typically achieved with populations isolated from the wild for many generations, caution is still needed when extrapolating this to field situations.

**Abstract:**

Discrepancies in the published research as to the attraction of the economically important pest western flower thrips (WFT) to different colours confounds the optimisation of field traps for pest management purposes. We considered whether the different experimental conditions of independent studies could have contributed to this. Therefore, the behavioural response (i.e., landings) to different colour cues of two WFT laboratory populations from Germany (DE) and The Netherlands (NL), which had previously been independently shown to have different colour preferences, were tested in the same place, and under the same experimental conditions. Single-choice wind tunnel bioassays supported previous independent findings, with more of a NL population landing on the yellow LED lamp (588 nm) than the blue (470 nm) (*p* = 0.022), and a not-statistically significant trend observed in a DE population landing more on blue compared to yellow (*p* = 0.104). To account for potential original host rearing influences, both populations were subsequently established on bean for ~20 weeks, then yellow chrysanthemum for 4–8 and 12–14 weeks and tested in wind tunnel choice bioassays. Laboratory of origin, irrespective of the host plant rearing regime, remained a significant effect (*p* < 0.001), with 65% of the NL WFT landing on yellow compared to blue (35%), while 66% of the DE WFT landed on blue compared to yellow (34%). There was also a significant host plant effect (*p* < 0.001), with increased response to yellow independent of laboratory of origin after rearing on chrysanthemum for 12–14 weeks. Results suggest that differing responses of WFT populations to colour is, in this case, independent of the experimental situation. Long-term separate isolation from the wild cannot be excluded as a cause, and the implications of this for optimising the trap colour is discussed.

## 1. Introduction

The western flower thrips (WFT) (*Frankliniella occidentalis* Pergande) is an economically important pest insect with a worldwide distribution affecting numerous horticultural crops. It causes direct cosmetic damage to flowers and fruits and is an important vector of tospoviruses in plants, which can lead to significant yield losses. Due to the pesticide resistance of WFT [1,2,3] and a stricter regulation of pesticides over environmental and human health concerns [4,5,6], more sustainable practices for pest control in agricultural production are desirable [6,7,8]. Colour is commonly used to attract WFT to the sticky traps to monitor populations [9,10], sometimes in combinations with odorant semiochemicals [11]. However, a better understanding of the attraction to colour could enhance captures for earlier detection and optimal timing of insecticide sprays as well as improve integrated pest management practices through mass trapping, lure and kill, or lure and infect tools [9,12]. Accordingly, although the effectiveness of traps for WFT has been explored extensively with the use of semiochemicals [11], more recently, the attraction of WFT to coloured light-emitting diodes (LEDs) [13,14,15,16] has been explored. For scientific research and for more a precise evaluation of thrips colour attraction, the use of LEDs has some advantages over otherwise coloured traps. In particular, LEDs do not rely on the reflection of ambient illumination and the intensity of the lights can be accurately controlled to remove any confounding interactions between the wavelength composition and light intensity. Additionally, LEDs emit narrower spectral curves compared to broader curves reflected by the coloured or painted surfaces used as traps, which allows for more specific wavelengths to be examined.

Despite multiple independent field and laboratory behavioural studies using coloured sticky traps or LEDs, plus continued widespread use of blue and yellow traps for pest management, there are still inconsistencies in the literature regarding the most effective colour for catching pest species of thrips in general [13,17]. For WFT, various studies have reported that blue is the most attractive colour [15,18,19,20,21], although other studies report that it is yellow [16,22,23,24,25] and some have shown differences in colour preference within the same study [26,27,28]. This apparent difference for blue and yellow is confounded by the range of extrinsic and intrinsic factors used in these different studies, which are known to have an effect on the behaviour of arthropods [29,30] including thrips [14,31,32,33,34]. Amongst these factors, the most obvious relates to methodological differences such as the rearing conditions, the preparation of insects prior to experiments, different experimental apparatus, and different conditions such as temperature, illumination, time of evaluation, duration of experiment, etc. Even the type of glue used on traps can influence the blue and yellow preference of WFT [35]. However, there are currently no studies that have considered such differences in the methodology as the cause for varying conclusions as to WFT attraction to blue and yellow. To test this, we used the same experimental design, conditions, and place to examine the responses to colour of two WFT populations recorded elsewhere as having different levels of attraction to blue and yellow. 

Two laboratory WFT populations were tested here, reared at Wageningen University and Research (Wageningen, The Netherlands), which had previously shown a higher attraction to yellow compared to other colours including blue [35], and Leibniz University Hannover (Hannover, Germany), which had independently shown a clear preference for blue compared to other colour cues including yellow [15,20]. The main objective was to observe these two populations in the same wind tunnel facility, under the same experimental conditions and host plant rearing regime to test the hypothesis that the differing response of these populations is independent of the experimental protocol. We also examined whether changing the host plant species had an influence on their colour preference. If differences in the behavioural responses to blue and yellow colours were reduced, this might infer that previous discrepancies were a consequence of different methodologies as opposed to biological intraspecific variation. In this way, alternative hypotheses may then be developed to advance our understanding of factors underlying the colour preference of WFT.

## 2. Materials and Methods

### 2.1. Insects: Management and Rearing of Colonies from The Netherlands and Germany

Base WFT populations were obtained from two different laboratory colonies: Wageningen University and Research (WUR), The Netherlands “NL” and the Institute of Horticultural Production Systems (Section of Phytomedicine), Leibniz Universität Hannover (LUH) “DE”. These were used in a molecular test for species status. All laboratory colonies derived from these two base WFT populations “DE” and “NL” were annotated according to the country of the institution from which they were obtained. All colonies were reared at WUR (Biointeractions and Plant Health Department) for the duration of this study. They were strictly isolated from each other in different climate chambers and rooms but with the same environmental conditions of 25–26 °C, 50–70% relative humidity (RH), and 16:8 light:dark period with lighting provided with Philips GreenPower LED Interlighting production module (Philips Electronics, Eindhoven, The Netherlands).

The Dutch base colony (NL) had been reared for at least 15 years, mostly on bean pods (*Phaseolus vulgaris* L.) to maintain a subsistence population and then occasionally on outsourced chrysanthemum potted plants (*Chrysanthemum morifolium* Ramat.) to increase the numbers for experimental work when needed. This was the same colony used by van Tol et al. [35] (see Figure 5 in that paper). A derived colony NL*c* was established from NL by rearing on yellow-flowered chrysanthemum potted plants (white-flowered plants were occasionally used if yellow could not be provided) for ~52 weeks and tested for attraction as a single-choice for a range of colours (*Experiment i*) (Figure 1). 

The base German colony (DE) had been reared for approximately 20 years, mostly on whole bean plants (*P. vulgaris* L.). After receiving a small sample of this population at WUR (25 September 2018), the colony DE*c* was derived from it (Figure 1) and reared on yellow chrysanthemum plants for ~6 weeks. Logistical issues prevented an assessment of the DE*c* at 52 weeks and ~6 weeks was the minimum time needed to build up the colony at WUR to establish its response to blue and yellow in a single-choice colour preference test (*Experiment ii*). 

For each NL and DE base colony, a further three derived colonies were established on different host-plant rearing regimes for comparison in the colour choice tests (*Experiment iii*). Colonies NL*b* and DE*b* were established from NL and DE, respectively, and reared on fresh green bean pods for at least ~20 weeks. The NL*bc*_4-8_ and DE*bc*_4-8_ colonies were established from NL*b* and DE*b*, respectively, and reared on chrysanthemum for 4 to 8 weeks. Finally, colonies NL*bc*_12-14_ and DE*bc*_12-14_ were a continuation of NL*bc*_4-8_ and DE*bc*_4-8_*,* respectively, and continued to be reared on chrysanthemum for 12 to 14 weeks (Figure 1). 

Reflectance spectra of the chrysanthemum flowers and bean pods used for the rearing of all colonies can be found in Appendix A.

### 2.2. DNA Analysis

To confirm the species status of the WFT colonies, standard cytochrome oxidase subunit I (COI) barcoding was undertaken for NL and DE as well as the other populations for an ‘evolutionary’ context. The populations for evolutionary analysis were collected in New Zealand, which included the New Zealand greenhouse WFT and biological strain variant New Zealand “Lupin” WFT [36], in addition to closely related species outgroups of *Frankliniella intonsa*, and *Thrips tabaci* (Onion Thrips, OT) *(*Appendix A). Three specimens from each of the six thrips taxa described above were used. Specimens were stored in 96% EtoH or propylene glycol at ~4 °C until DNA extraction. Total DNA was extracted from whole single insects based on a standardised method [37] (Appendix 1 therein), using the Animal Tissue protocol from the DNeasy Blood and Tissue Kit (Qiagen, Venlo, The Netherlands) and final elution with 100 µL of elution buffer pre-heated to 56 °C to increase the final DNA concentration. PCR amplification and sequencing were carried out using the method from OEPP/EPPO [37] (Appendix 1 therein) with PCR primers LCO1490 and HCO2198 [38]. PCR amplification followed the manufacturer’s instructions for the Bio-X-ACT Short mix (BiolinePty. Ltd., Sydney, Australia) with a PCR reaction volume of 25 µL including 5 µL of the DNA template. The PCR cycle was 3 min at 94 °C, followed by 40 cycles of 30 s at 94 °C, 1 min at 51 °C, 1 min at 72 °C, and a final extension of 10 min at 72 °C. Sanger sequencing was carried out with BigDye™ Terminator v3.1 Cycle Sequencing Kit and the 3500XL Genetic Analyzer (Applied Biosystems, Thermo Fisher Scientific, Waltham, MA, USA). 

Consensus sequences for each specimen were generated by aligning the forward and reverse sequences using the MUSCLE algorithm in MEGA X [39]. All sequences obtained were deposited in the NCBI GenBank under accession numbers ON310562–ON310579; populations from which those specimens were taken are vouchered in the Lincoln University Entomology Research Collection. Additionally, 76 DNA sequences for the same COI region of closely related species *Frankliniella occidentalis*, *F. borinquen*, *F. intonsa*, and *Thrips tabaci* (as the outgroup) were randomly selected and downloaded from BOLD (https://www.boldsystems.org/; accessed on 5 March 2021) *(*Appendix A). All sequences were used in a neighbour-joining analysis in MEGA X [39] using the pair-wise Kimura-2-parameter (K2P) model [40] and a bootstrap of 10,000 replicates. Genetic distances among the taxa were visualised in a neighbour-joining tree as a summary of intra- and inter-specific diversity. Species identification of each of the 18 specimens of six different thrips samples was conducted using the BOLD Identification System (IDS) (https://www.boldsystems.org/; accessed on 5 March 2021), which uses >5000 thrips barcode sequences for comparison. 

### 2.3. Wind Tunnel Setup

The experimental arrangement was similar to that described in van Tol et al. [35]. All experiments were conducted at the WUR facility in a wind tunnel 3.0 m (l) × 1.30 m (w) × 0.7 m (h) made of transparent glass (Figure 2). The test arena was located in the middle of the wind tunnel and the ceiling above it was covered with a transparent polyethylene diffusing sheet (Suncover Nectarine C-980, Ginegar Plastic Products Ltd., Ginnegar, Israel). The floor of the three compartments was black. Wind speed inside the wind tunnel was set at 0.3 m/s, temperature at 26 °C, and RH at 65–70%.

The LED lamp, used as the main colour stimulus for the thrips in the wind tunnel, consisted of a 3D-printed black dome (Figure 3) with a 18.5 cm diameter covered inside with aluminium foil to increase the light reflection inside the lamp with a division in the dome that effectively partitioned the lamp in half (Figure 3A). A light diffusing glass plate (Edmund Optics Ltd., York, UK) was glued at the distal part of the lamp with an effective area of 103.86 cm^2^ where thrips could land (Figure 3C). For each trial, a thin layer of sticky glue P300I35 (Intercol Industrial adhesives, Ede, The Netherlands) was evenly spread on one side of a 100 µm transparent polypropylene sheet (Staples Solutions, Amsterdam, The Netherlands), which was placed on the exterior of the glass plate and taped to the removable frame of the lamp (Figure 3C,D). The LEDs were placed in the back of the lamp (Figure 3E) and positioned so that light would shine forward through the diffusing glass plate. Up to two coloured LED lights could be housed in the dome at the same time (Figure 3B,F), which could be the same or a different colour. When only one or two LEDs were used, the holes without LEDs were covered with cork to avoid light escaping. The lamp was connected to a clamp and positioned at an angle of 45° in relation to the wind-tunnel floor, facing down on the test arena. The centre of the lamp was fixed at 60 cm from the floor and the release platform (height 8 cm) was placed 56 cm away from the base of the lamp. 

General lighting in the wind tunnel was provided by LED strips situated on the ceiling over the test arena (Figure 2, black arrows). This had a light spectrum component in the visible range 400–750 nm (LED-strip–Full-colour RGB + Warm White—24V High Power Protected 5050, LuxaLight, Eindhoven, The Netherlands) and a component in the ultraviolet light (UV-A) range from 360 to 390 nm (strip of UV LED Engin, LZ4-04UV00, Osram Sylvania Inc., Wilmington, MA, USA) (Appendix A). The amount of UV-A emitted from the ceiling LEDs was set as 2% of the visible light to approximate that found in natural sunlight.

The emission spectra and intensity for the LED ceiling strips and lamp stimulus were measured and adjusted inside the wind tunnel using a broadband spectroradiometer Specbos 1211UV (JETI Technische Instrumente GmbH, Jena, Germany). The intensity of the light from the lamp was controlled by manipulating the amperes in the power controllers connected to the LEDs. All LEDs used in the lamp were set to a spectral radiance of 9.0 × 10^17^ photons·s^−1^·sr^−1^·m^−2^ (i.e., all LEDs emitted the same amount of light). No difference in LED spectral radiance was found with or without the polypropylene sheet and glue that were placed on the lamp to catch the WFT when measured with the spectroradiometer. 

### 2.4. Experiments with LED Colours

Adult females (distinguished from males by their larger size) of unknown age and mating status were removed from the relevant colony by an aspirator, sedated with CO_2_ for 15–20 s, and then 100 were transferred to a plastic transparent container (‘release container’) with the dimensions of 7 cm (h) × 7 cm (d). The release container was covered on the top with Parafilm (Bemis, Neenah, WI, USA) and placed on the release platform for 1 h before starting the experiments without water or food for the thrips. Each replicate tested the response of 100 insects. The thrips were released simultaneously into the tunnel from the release platform by removing the Parafilm with minimal physical disturbance. Observations were made on the position of the thrips in the test arena after 6 h. Several replicates were conducted per experiment, but only one 6 h trial was undertaken per day to ensure that the wind tunnel was clear of any WFT remaining from the previous trial. The experiments conducted were: 


**
*Experiment i:*
**
*Single-choice response of NLc to different LED colours.*


Seven different LED spectra were presented to the thrips in separate trials (Table 1). The spectral range spanned the UV and visible light from 360 to 700 nm (Appendix A, Appendix A; the spectral sensitivity of WFT shown in Appendix A). Data from the treatments evaluating the higher light intensities were included in this experiment (data not shown as no statistically significant differences were observed), but all data were combined and included in the final statistical analysis to consider the effect of colour. Four replicate runs of each light spectrum were carried out. The running order of the treatments and replicates was generated with CycDesign 5.1 (VSN International Ltd. 2013, Hemel Hempstead, UK) using a Latinised resolvable block design to distribute replicates of treatments in “blocks” for the even distribution of treatments over time. Each block comprised six days. The number of thrips trapped on the glass plate of the LED lamp and the ones remaining in the release container (dead and alive) were recorded. 


**
*Experiment ii:*
**
*Single-choice response of DEc to blue and yellow LEDs.*


This experiment was conducted to assess whether the previously observed preference for blue of the DE population [15,20] was reproduced at the WUR facilities. The same experimental setup and conditions as described in *Experiment i* were used, but only the colours blue (477 nm) or yellow (588 nm) (Appendix A) were presented to the thrips as a single treatment, with four replicates per treatment. Treatments were run in a systematic way, alternating between the colours. The number of thrips caught on the glass plate of the LED lamp and the ones remaining inside the release container (dead and alive) were recorded at the end of each replicate. 


**
*Experiment iii:*
**
*Choice-test response to blue and yellow by NL and DE WFT from different host plant rearing regimes.*


To test the potential influence of immediate colony history as an aspect beyond experimental setup on the attraction toward blue and yellow, a choice bioassay examined the effects of the laboratory of origin and host-plant rearing regime using the colonies derived from NL and DE (Figure 1). In contrast to *Experiments i* and *ii*, the blue (477 nm) and yellow (588 nm) LEDs were presented at the same time, adjacent to each other in the lamp. Six derived colonies were tested, being those reared on bean pods (NL*b* and DE*b*), transferred from bean to chrysanthemum for 4–8 weeks (NL*bc*_4-8_ and DE*bc*_4-8_), and transferred from bean to chrysanthemum for 12–14 weeks (NL*bc*_12-14_ and DE*bc*_12-14_) (Figure 1). Each of the six treatments was replicated six times. The running order of the WFT from both countries was partly randomised. Practical necessity meant that thrips from NL*b* and DE*b* were run over two blocks of days, whilst all of the replicates for NL*bc*_4-8_ and DE*bc*_4-8_ were run in a single block, and similarly NL*bc*_12-14_ and DE*bc*_12-14_ were also run in a single block. The left and right position of the colours was alternated between replicates. Light spectra of the blue and yellow LEDs used are illustrated in Appendix A. 

After 6 h, the number of WFT were counted and included in five categories: either (1) remaining inside the *release container* (dead or alive), (2) on the *ceiling* under the UV-A illumination, or having landed on (3) the *blue* or (4) the *yellow* sides of the sticky glass plate. All thrips that flew out of the release container but did not respond to any light stimuli from the previous categories were assumed to be (5) *elsewhere* inside the wind tunnel. 

### 2.5. Data Analysis

***Experiment i and ii:*** Percentages were calculated for insects caught on the glass plate of the LED lamp out of the 100 released for each treatment and replicate, and the mean percentages calculated per treatment. For both *Experiments i* and *ii*, the number of thrips caught on the sticky glass plate after six hours, out of the total number of thrips that were released inside the wind tunnel, was analysed using a binomial generalised linear model [41] with a logit link function and the dispersion was estimated. Because there was a median of only one dead thrips per run, the number of WFT released (100) was used as the binomial total, not adjusting for dead WFT. Treatment effects were assessed using F-tests. The percentage of thrips on the sticky glass plate and associated 95% confidence limits were calculated from predictions made on the link (logit) scale, and back-transformed for presentation. All analyses were carried out with Genstat [42].

***Experiment iii:*** Initial data analyses (not shown) showed no significant evidence of a difference between the left or right position of the blue and yellow LEDs (*p* > 0.05). Therefore, the data were analysed ignoring the LED side position. Because the proportion of thrips *dead* or remaining in the *release container* did not vary strongly between treatments (*p* > 0.2 and *p* > 0.3, respectively), these categories were ignored in the subsequent analyses. Data were analysed in two ways, referred to as ‘*analysis a’* and ‘*analysis b’* here after.



*Analysis a: WFT that landed on blue or yellow on the glass plate of the LED lamp.*



This analysis focused simply on the attraction to the lamp as reported in other studies (see Discussion), considering only the percentage of thrips that landed on *blue* or *yellow* only, out of the total that landed on both colours (see Section 2.4, Experiment iii—sum of categories (3) and (4)). Data were analysed using a binomial generalised linear model (GLM) [41] with a logit link, the dispersion estimated, and the total on the sticky glass plate as the binomial total. The effect of host rearing regime, laboratory of origin, and the interaction between them were assessed using F-tests for both analyses. The estimated percentages of thrips on each light colour and their associated 95% confidence intervals (CI) were obtained on the link scale and back-transformed for presentation.



*Analysis b: Response of WFT relative to the thrips that left the release container.*



This analysis was carried out to include thrips that did not land on the sticky glass plate but flew out of the release container. Thus, the data were analysed as the percentage of thrips at each of the four remaining categories ((2) *blue*, (3) *yellow*, (4) *ceiling*, and (5) *elsewhere*, see Section 2.4, *Experiment iii*) out of the total number of thrips that left the release container. In this analysis, because there were more than two categories included, the data were analysed using a Poisson log-linear model for multinomial data [41] with the dispersion estimated. The estimated percentages and associated confidence limits were obtained using separate binomial analysis for each of the categories, similarly to the binomial analysis above, but using the estimated dispersion from the multinomial analysis.

## 3. Results

### 3.1. Species Status of the Dutch (NL) and German (DE) Colonies

The DNA barcode identification for the NL and DE specimens showed a 100% match to other *F. occidentalis* in the large thrips dataset (Appendix A) in the BOLD Identification System (IDS). Furthermore, the neighbour-joining analysis of 18 new sequences generated in this study, together with the 76 selected independently published sequences, confirmed that the NL and DE populations used for the experiments in this study clearly fall into the *F. occidentalis* clade (Appendix A). Consistent with this, the ~1.18% genetic distance between the NL and DE colonies (Appendix A) does not support them as different species based on the >10-fold genetic distances between *F. occidentalis* and other species within the genus such as *F. borinquen* (16%) and *F. intonsa* (23%) (Appendix A). The distance between the biological “glasshouse” and “lupin” strains of *F. occidentalis* was ~4%, which is greater than between the NL and DE *F. occidentalis* (Appendix A). Genetic variation within the DE colony was lower (0.12%) than that within the NL colony (2.08%). 

### 3.2. Experiment i: Response of NLc to Different LED Colours

In a single-choice test, the percentage of NL WFT reared on chrysanthemum that were caught on the sticky glass plate of the LED lamp varied significantly with the light colour (*p* < 0.001) (Figure 4). The two most preferred colours were UV-A and yellow. UV-A light attracted the highest mean percentage with 37.3%, closely followed by yellow with a mean of 35.5% (*p* = 0.789 compared to UV). Blue was the third most attractive colour, with a mean of 21%, but this was significantly lower than the WFT caught by UV-A and yellow (*p* = 0.011 and 0.022, respectively). The rest of the colours evaluated attracted similar mean percentages of WFT to each other (violet at 12.8%, red 10.8%, cyan 7.5% and green 7.3%), which were considerably lower than that of UV-A, yellow, or blue (Appendix A).

### 3.3. Experiment ii: Response of DEc to Blue and Yellow LEDs

In a single-choice test, the DE WFT reared on chrysanthemum showed a trend towards blue over yellow, with 34% caught on blue compared to 24% on yellow (Figure 5), although the difference was not statistically significant (*p* = 0.104). 

### 3.4. Experiment iii: Blue and Yellow LED Choice Test of Dutch (NL) and German (DE) WFT from Different Host Plant Rearing Regimes 

The influence of the laboratory of origin and immediate host history, and host rearing regime were evaluated by observing (a) how the numbers of WFT caught were divided between the yellow and blue halves of the lamp, and (b) that number relative to the insects that left the release container and were registered in other places inside the wind tunnel. 

#### 3.4.1. Analysis a: Response of WFT that Landed on Blue or Yellow on the Glass Plate of the LED Lamp

Considering only those WFT that had been attracted to the lamp, there was a statistically significant variation between the countries of origin and rearing regimes (*p* < 0.001 for both the laboratory of origin and rearing regime the as main effects) for the percentage that were caught on the *yellow* or *blue* half of the lamp, with a similar difference between rearing on different host plants for the laboratory of origin (*p* = 0.118 for the interaction) (Figure 6). Related to the laboratory of origin, on average, a higher percentage of NL thrips landed on yellow (65%) compared to the DE thrips (34%), while a higher percentage of DE thrips landed on blue (66%) compared to the NL thrips (35%). This difference remained even after rearing on different host plants, with the exception that NL*b* showed no clear preference to blue or yellow. Related to host rearing regime, on average, a lower proportion of thrips that had been reared on bean (NL*b*) and DE*b*) landed on yellow (39%) compared to those that landed on blue (61%). In contrast, for the colonies reared on chrysanthemum for 4–8 weeks (NL*bc*_4-8_ and DE*bc*_4-8_), on average, landings on yellow (61%) were higher compared to landings on blue (39%). This trend was maintained for the 12–14 weeks of chrysanthemum rearing (NL*bc*_12-14_ and DE*bc*_12-14_), with a higher average of WFT landing on yellow (56%) compared to blue (44%). Effectively, the average landings of thrips on yellow from both the NL and DE colonies increased when reared on chrysanthemum after being reared on bean. 

#### 3.4.2. *Analysis b*: *Response of WFT Relative to the Thrips That Left the Release Container*

In addition to WFT landing on the *blue* or *yellow* halves of the sticky glass plate, the distributions of those that left the release container were also recorded as either on the *ceiling* (UV-A illuminated section) or *elsewhere*. The overall pattern of the distribution of thrips across the four locations varied significantly between the laboratory of origin (*p* < 0.001) as well as rearing regime (*p* = 0.001) with a statistically significant interaction between the effects (*p* = 0.004 for the interaction) (Figure 7).

As for *analysis a*, irrespective of the host rearing regime, the percentage of WFT landing on *yellow* was greater on average for the NL (21%) compared to the DE WFT (8%) (*p* < 0.001). For all thrips landing on *yellow*, the interaction between the laboratory of origin and rearing regime was not significant (*p* = 0.803), reflecting the similar landing patterns observed for the three rearing regimes for the two countries of origin. However, in contrast to *analysis a*, the percentage of landings on *blue* did not vary significantly between the laboratory of origin, with 11% of the NL thrips and 15% of the DE thrips (*p* = 0.124).

Similarly, as for *analysis a*, by the host rearing regime irrespective of the laboratory of origin, the average percentage of WFT that landed on *yellow* was lowest for the thrips reared on bean, (10%) but increased to 15% when reared on chrysanthemum after 4–8 weeks and to 18% after 12–14 weeks (*p* = 0.021). In contrast, for landings on *blue*, a trend with the host rearing regime was not observed in *analysis b* or in analysis *a*, with 15% when reared on bean, 10% on chrysanthemum for 4-8 weeks, and 14% on chrysanthemum for 12–14 weeks (*p* = 0.142; *p* = 0.178 for the interaction). 

For those attracted to the UV-A light from the *ceiling,* the percentage varied with both laboratory of origin and host rearing regime (*p* < 0.001 for the interaction). The mean percentage of NL WFT that landed on the *ceiling* varied substantially between rearing regimes with only 3.9% from those on bean (NL*b*) compared to 26.5% and 17.2% from chrysanthemum (NL*bc*_4-8_ and NL*bc*_12-14_, respectively). For the DE WFT, there was no such host correlation with the mean percentage landing on the *ceiling*, with those from bean (DE*b*) at 23.4% and those from chrysanthemum (DE*bc*_4-8_ and DE*bc*_12-14_) at 23.9% and 22.6%, respectively. 

For the thrips considered to be *elsewhere* in the wind tunnel, the number varied little between the laboratory of origin (*p* = 0.523) but was significantly different amongst the host rearing regimes (*p* = 0.004 for the main effect, *p* = 0.151 for the interaction between the laboratory and rearing regime). More of those reared on bean (NL*b* and DE*b*) were *elsewhere* (mean of 62.5%) compared to the two chrysanthemum regimes of 50% (NL*bc*_4-8_ and DE*bc*_4-8_) and 48% (NL*bc*_12-14_ and DE*bc*_12-14_) (Figure 7).

For the thrips considered to be *elsewhere* in the wind tunnel, the number varied little between the laboratory of origin (*p* = 0.523) but was significantly different amongst the host rearing regimes (*p* = 0.004 for the main effect, *p* = 0.151 for the interaction between the laboratory and rearing regime). More of those reared on bean (NL*b* and DE*b*) were *elsewhere* (mean of 62.5%) compared to the two chrysanthemum regimes of 50% (NL*bc*_4-8_ and DE*bc*_4-8_) and 48% (NL*bc*_12-14_ and DE*bc*_12-14_) (Figure 7).

## 4. Discussion

Observing the behaviour of the two different WFT populations in the same laboratory and under the same conditions allowed us to examine whether within-species discrepancy in colour preference, recorded amongst separate studies [15,20,35], was purely a function of methodology. The data presented in our study did in fact indicate that methodology alone could not account for the consistently different responses, in particular to blue and yellow, as each population maintained its original colour preference when evaluated under the same conditions. Specifically, with the exception of NL thrips reared on beans (NL*b*), the NL population was on average more attracted to yellow compared to blue, in either a single-choice situation (*Experiment i*), or with a two-choice scenario (*Experiment iii*), as also observed in prior bioassays using coloured sticky plates by van Tol et al. [35]. On the other hand, the DE population had a stronger preference for blue over yellow (*Experiments ii* and *iii*), as reported in bioassays using LEDs with thrips taken from the source laboratory by Otieno et al. [15] and Stukenberg et al. [20]. 

Our results demonstrate for the first time that two different populations of WFT accessed from independent laboratories showed a different colour preference, even when evaluated under the same experimental setup. However, beyond evaluating the use of a standardised setup, the manipulation of rearing regime indicated that the host plant species could influence this yellow/blue attraction (*Experiment iii*). Factors such as genetics and previous experience might therefore have impacted on the response of the populations to colour, both of which are known to influence the behaviour of insects towards sensory stimuli in general [29,30], but little attention has been paid to these aspects in the case of colour, especially in small insects such as thrips. 

To account for any potentially species-level genetic difference, using COI barcoding [36,43,44], our molecular data indicated that the NL and DE populations were both *F. occidentalis*. The divergence between them at 1.18% was consistent with intraspecific variation in WFT reported elsewhere typically as 2–3% [43,45], while inter-species divergences of closely related species of the genus *Frankliniella* and other thrips genera were generally around 10–21% [43,45,46], although it can be as low as 4.4% for cryptic species like some of those found in the genus *Pseudophilothrips* [46]. Caution about species status was also driven by the evidence building for *F. occidentalis* as a cryptic species complex, revealing clades for “glasshouse” and “lupin” through COI barcoding [36,45], which exhibit reproductive and developmental differences and cause varying levels of crop damage [47]. Sub-specific genetic differences, which may or may not have resulted in the observed behavioural responses of these two populations, is nonetheless possible. The colonies evaluated in this study likely originated from a different gene pool and have been kept geographically isolated from each other for more than two decades. The populations could therefore continue to adapt to their local rearing conditions and diverge independently. Much more rapidly evolving markers would need to be employed to explore this further. For example, restricted gene flow between populations has been proposed as the reason for genetic differences among geographically distinct WFT populations [48,49]. Laboratory “strains” of WFT through physical and temporal isolation have been suggested to develop and present differences in the feeding damage and reproductive success compared to wild WFT “greenhouse” strain populations from around The Netherlands within the short time frame of ~100 generations [49]. However, intra-specific genetic differences based on microsatellite data have been associated with extrinsic habitat-related factors such as temperature and precipitation, and not necessarily geographic distance or origin [50]. In the current study, if any genetic differences do exist between the two populations, they could have arisen as a result of laboratory adaptation after more than two decades (>150 generations) of being reared in laboratory conditions, or host-related epigenetic inheritance, or even by chance, as has been proposed for behavioural intraspecific variation observed in other insects [30,51]. 

For the aspect of our study that considered differences between populations and the host plant (*Experiment iii*), we also took the opportunity to present two analyses of the same data. First, to make the data more comparable with the common practice elsewhere of not presenting the data of insects that did not respond to the stimuli in choice studies in a flight cage [14,15,16,20,35], *analysis a* delimited evaluation to only the proportion of WFT that landed on the blue or yellow side of the sticky glass plate. This confirmed the general preference difference established between colonies and that the tendency for yellow was improved with rearing on yellow chrysanthemum; it did not indicate whether blue just became less attractive in the presence of yellow after rearing on chrysanthemum. However, in *analysis b,* observing all WFT that flew and left the release container in the wind tunnel showed that the proportion landing on blue remained similar, regardless of the laboratory of origin or rearing regime. The results instead suggest that chrysanthemum increased responsiveness to yellow in both populations, drawing in thrips from those that were otherwise *elsewhere* in the wind tunnel as opposed to reducing the attractiveness of blue or UV-A per se. This was not possible to be concluded from *analysis a*. One other potential explanation for the increased response to yellow from both populations is that it was a consequence of the unintended introduction of wild WFT from the organically grown chrysanthemum brought to the laboratory for the rearing. However, this has been discounted as being extremely unlikely. Not only is there is no evidence to suggest that wild populations of WFT from The Netherlands have a unique relative preference of yellow over blue [35], but any thrips present were undetectable through standard inspection of the flowers on arrival; at best, the numbers would have been too low to have genetically impacted on the colonies over the short number of generations (~3–4) that thrips were put on chrysanthemum for this experiment. 

The increased response to yellow observed in *Experiment iii* by the varying rearing regimes suggests an influence of previous host experience [52]. Colour is thought to play an important role in host-searching behaviour in thrips as well as other groups of insects [53,54,55,56,57,58]. Associative learning is also common in insect species that strongly depend on flowers for food [59,60,61,62,63,64]. Even though WFT is highly polyphagous [65], the presence of flowers [66] and flower colour are important [58] for their ecology and reproductive success. Additionally, WFT can show a preference and perform differently on certain types of flowering plants or specific plant species [49,67,68] including those that have yellow flowers [69] or that provide pollen [70]. Therefore, for both populations, the increased attraction to yellow after the shift from beans to yellow-flowered chrysanthemum could potentially be explained as WFT perhaps being able to associate colour with food reward. Certainly, yellow chrysanthemum flowers are highly suitable as a food source for thrips [71,72,73,74,75], probably due to pollen being an important supplement to their diet for fitness and reproduction success [55,70,76,77,78,79,80], and which is lacking with the vegetative bean pods. 

The thrips species *Frankliniella schulzei* has been inferred to make this connection between colour and host by adult females showing a strong wavelength-specific response to red [81], the same colour as the flowers of their postulated ‘primary host’ plant *Malvaviscus arboreus* [81,82]. In other insect orders, different populations of the same species can also exhibit affinities for a different colour such as the Sardinian Island population of *Bombus terrestris sassaricus,* which showed a preference for artificial red flowers in choice tests, compared to the mainland populations of *B. terrestris* that only showed a preference towards violet and the blue range of the light spectrum [51,83]. 

Associative learning has not been studied in Thysanoptera, but it has been found in a variety of orders of insects where it has been proposed that any animal with a nervous system has the potential to learn [84]. Extremely small insects with very small brains such as the featherwing beetle (*Nephanes titan*) (Coleoptera: Ptiilidae) have been found to learn to associate colour with food [85], showing that brain size is not an impediment for learning [86]. However, whether species of Thysanoptera are capable of shifting their colour preference based on the colour of host plant species remains a hypothesis for further study. Both blue and yellow are relevant colours for WFT and wavelengths in the blue and yellow range of the light spectrum are both found in flowers in nature, but whether the resulting differences in preference of these two colours are a result of adaptation to environment, selection pressure, learning, and experience or mere chance is not yet understood. 

Although the focus of this study was to establish the population response to yellow and blue, on average, only around 20–35% of WFT released into the wind tunnel across all experiments landed on the colour stimuli. As shown in *analysis b* of *Experiment iii*, the majority of thrips did not respond to the light colours coming from the LED lamp and were captured either on ‘UV-A *ceiling’* or found ‘*elsewhere*’. The low number of responsive thrips could have been partly caused by the large size of the wind tunnel used. This would be consistent with the higher percentages of thrips attracted to light stimuli recorded in other studies that used much smaller apparatus [15,20], some using extremely confined space to evaluate thrips phototaxis [87]. Nonetheless, results with similar proportions of WFT landing on visual cues to those presented in this study have been found in comparable wind tunnel bioassays evaluating the attraction of WFT to colour and odour stimuli [22,31,35]. In our study, of those that did not land on the sticky glass plate but were “*elsewhere*” in the wind tunnel, a greater proportion had been reared on bean compared to chrysanthemum, although it is not clear why. Additionally, the UV-A *ceiling* light elicited a relatively strong response similar to that of the main blue and yellow cues. The exception was the low proportion of thrips from NL*b* that were attracted to the UV-A source, with no clear explanation as to why. From the preliminary experiments conducted with the UV-light off (unpublished data) in the same facilities and with the same setup, the percentage of NL*c* WFT attracted to the LED lights remained similar while the numbers of WFT found on the ceiling dropped considerably. Thus, it is thought that the response of WFT flying towards the ceiling was mainly caused by the presence of UV-A light and not by the white light illuminating the wind tunnel. UV-light is known to be an important cue in aiding invertebrates in navigation, orientation [88], and the dispersion of pest insects [89,90,91] including thrips [92]. It also elicits phototaxis in insects [93,94] and is thought to be related to wavelength-specific behaviours [88,93,94] such as migration or escape response [94]. Potentially, therefore, the proportion of WFT attracted to UV-A light in our experiments could have been in a different physiological state to those attracted to blue or yellow, whose response instead may have been elicited by hunger-related physiology, leading to host-seeking behaviour.

## 5. Conclusions

The most important finding here is that with consistent experimental methods, blue and yellow attraction was confirmed for the first time to vary significantly between different populations of WFT. This suggests the presence of underlying biological or environmental factors, which complicates the extrapolation and generalisation of laboratory data on the response of WFT to colour, especially to improve trapping in the field. Additionally, despite similar long-term maintenance on bean for both base colonies used here, recent host plant rearing on yellow chrysanthemum was shown to influence their relative colour preference. The mechanisms responsible for the differences found between the NL and DE evaluated in this study remain unclear and warrant further investigation. However, to understand whether either blue or yellow are inherent colour preferences of WFT, studies with strictly naïve WFT would be needed. Innate preferences towards wavelengths in the blue spectrum have been reported for other insects that have a strong association with flowers, but this may depend on the background colour or intensity in some cases [95,96,97,98,99,100,101]. Likewise, in other herbivorous insects, colour cues in the green-yellow part of the light spectrum can elicit strong behavioural responses that are possibly an innate preference and known as ‘wavelength specific behaviours’ [56,91,102,103]. In any event, the extrapolation of studies conducted on colour with established laboratory colonies for application for WFT trapping in the field needs to be carefully considered. Potentially, there are strains or biotypes of WFT that naturally vary in their attraction to certain colours, so that no one colour will be the most effective to help maximise trap catches. Therefore, the behavioural responses of local wild WFT populations should be considered for trapping and/or monitoring.

## Figures and Tables

**Figure 1 insects-13-00538-f001:**
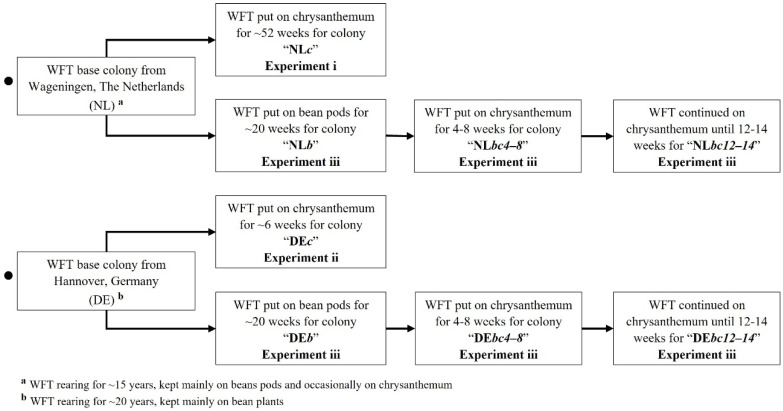
Diagram showing the derivations of WFT colonies from Wageningen, The Netherlands (NL) and Hannover, Germany (DE) from the original base colonies and used in Experiments i–iii.

**Figure 2 insects-13-00538-f002:**
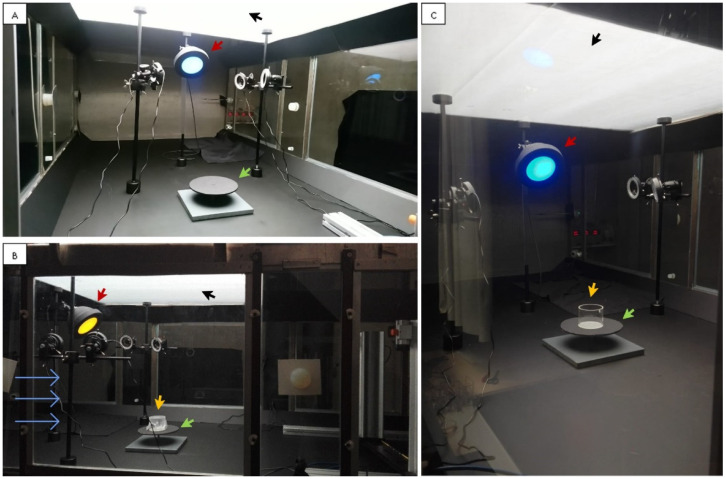
The wind tunnel setup used for the WFT behavioural experiments viewed from different perspectives illustrating: (**A**) The complete equipment arrangement; (**B**) the yellow stimulus, release container and wind direction; (**C**) the release container in relation to the LED stimulus (blue). Red arrow, LED lamp (dome); green arrow, insect release platform; yellow arrow, insect release container; blue arrows, wind direction inside the wind tunnel; black arrow, general ceiling illumination provided for the wind tunnel (includes wavelengths in the visible light spectrum and UV-A light).

**Figure 3 insects-13-00538-f003:**
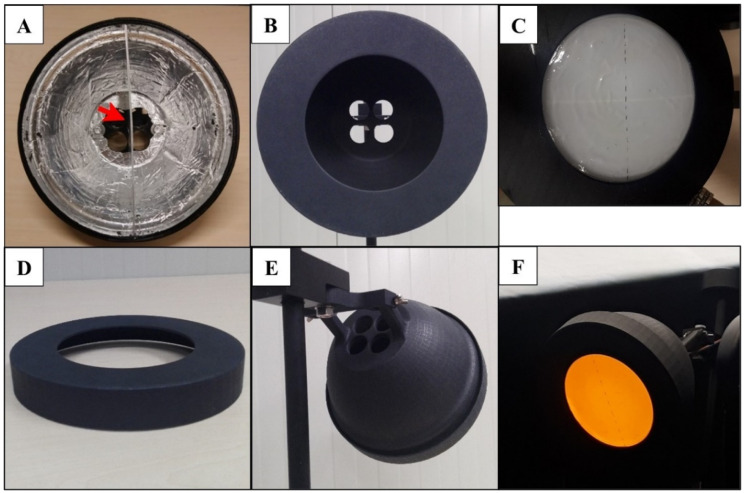
Aspects of the 3D printed dome housing the LEDs. (**A**) Frontal view showing the interior covered with aluminium foil, red arrow shows the division inside the dome. (**B**) Frontal view with the frame attached. (**C**) LED lamp with the diffusing glass plate and transparent propylene sheet with sticky glue. (**D**) Removable frame where the transparent sticky plastic was placed for catching the WFT. (**E**) Back side of the dome showing the four visible orifices where LEDs were placed. (**F**) LED lamp inside the wind tunnel shining a yellow and blue LED from inside the dome.

**Figure 4 insects-13-00538-f004:**
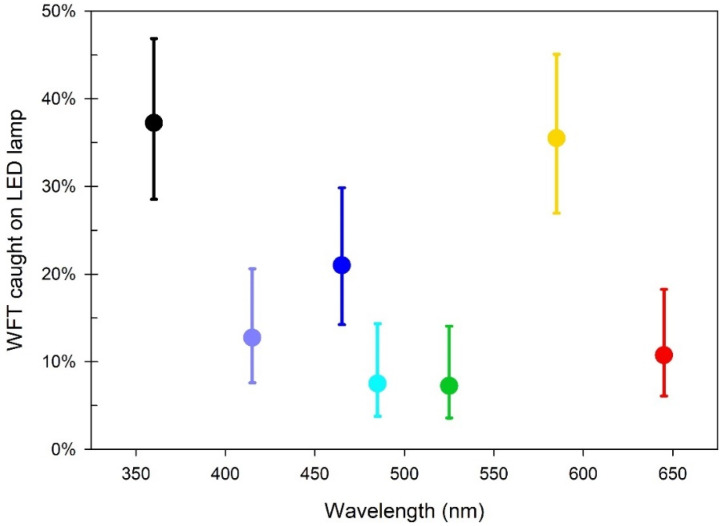
*Experiment i*: Mean percentage of Dutch WFT (NL*c*) caught on the sticky glass plate of the LED lamp with different colours six hours after release. N = 4 replicates. Error bars are 95% confidence intervals. Peak wavelength of LEDs used: Pink (UV-A 369 nm), Violet (422 nm), Blue (477 nm), Cyan (502 nm), Green (529 nm), Yellow (588 nm), Red (651 nm). Each LED was presented as a single choice. All information from this figure is provided in Appendix A.

**Figure 5 insects-13-00538-f005:**
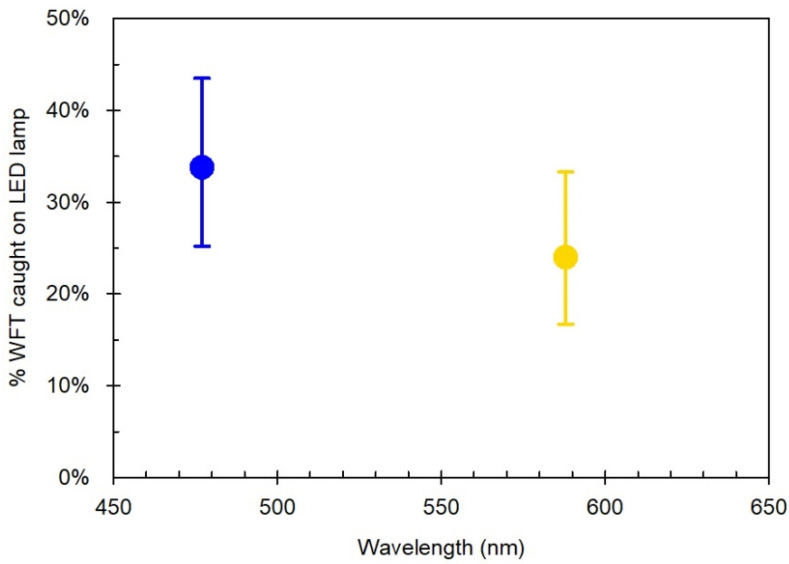
*Experiment ii*: Mean percentage of German WFT (DE*c*) caught on the sticky glass plate of the LED lamp with blue or yellow light six hours after the release of the insects. Error bars are 95% confidence intervals. Peak wavelength of LEDs: Blue (477 nm), Yellow (588 nm). Each LED was presented as a single choice. All information from this figure is provided in Appendix A.

**Figure 6 insects-13-00538-f006:**
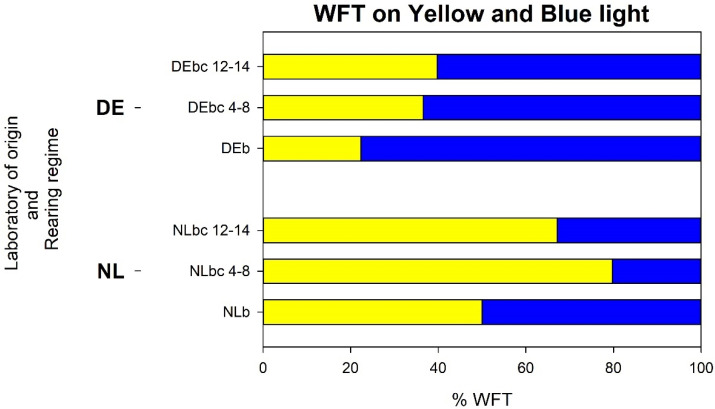
*Experiment iii, analysis a*: Stacked bar chart showing the percentages of WFT found on the yellow or blue light per laboratory of origin and rearing regime out of the total number of thrips that landed on the sticky glass plate of the LED lamp. Peak wavelength of LEDs: Blue (477 nm), Yellow (588 nm). DEb: colony of German WFT reared on bean pods for ~20 weeks; DEbc 4-8: DEb colony transferred and reared on yellow-flowered chrysanthemum plants for 4 to 8 weeks; DEbc 12-14: continuation of colony DEbc 4-8 until reaching 12 to 14 weeks reared on yellow-flowered chrysanthemum plants. NLb: colony of Dutch WFT reared on bean pods for ~20 weeks; NLbc 4-8: NLb colony transferred and reared on yellow-flowered chrysanthemum plants for 4 to 8 weeks; NLbc 12-14: continuation of colony NLbc4-8 until reaching 12 to 14 weeks reared on yellow-flowered chrysanthemum plants. All information from this figure including mean percentages and 95% confidence intervals is provided in Appendix A.

**Figure 7 insects-13-00538-f007:**
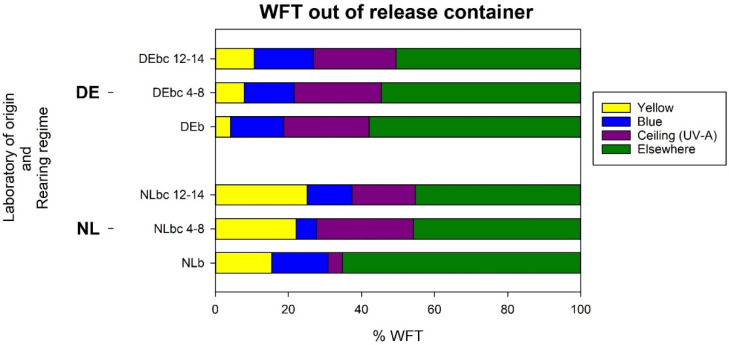
*Experiment iii, analysis b*: Stacked bar chart showing the percentages of WFT that left the release container as per the laboratory of origin and rearing regime for each of the four categories in which the WFT was grouped inside the wind tunnel. Peak wavelength of LEDs: Blue (477 nm), Yellow (588 nm). DEb: colony of German WFT reared on bean pods for ~20 weeks; DEbc 4-8: DEb colony transferred and reared on yellow-flowered chrysanthemum plants for 4 to 8 weeks; DEbc 12-14: continuation of colony DEbc 4-8 until reaching 12 to 14 weeks reared on yellow-flowered chrysanthemum plants. NLb: colony of Dutch WFT reared on bean pods for ~20 weeks; NLbc 4-8: NLb colony transferred and reared on yellow-flowered chrysanthemum plants for 4 to 8 weeks; NLbc 12-14: continuation of colony NLbc4-8 until reaching 12 to 14 weeks reared on yellow-flowered chrysanthemum plants. All information from this figure including mean percentages and 95% confidence intervals is provided in Appendix A.

**Table 1 insects-13-00538-t001:** The list of LED treatments to test for the colour preference of Dutch (NL*c*) WFT.

Treatment	LED Colour ^a^	Peak Wavelength (nm)
1	UV-A	369
2	Violet	422
3	Blue	477
4	Cyan	502
5	Green	529
6	Yellow	588
7	Red	651

^a^ The spectral radiance of 9.0 × 10^17^ photons·s^−1^·sr^−1^·m^−2^.

## Data Availability

The data presented in this study are available on request from the corresponding author.

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
