# Peer review of "Colour Response in Western Flower Thrips Varies Intraspecifically"

_insects, 2022, doi:10.3390/insects13060538_

Round 1

Reviewer 1 Report

This paper seeks to ratify previous works showing different colour preferences of WFT under a set of standard conditions. Differences in preference of Dutch and German populations were confirmed and evidence provided for an effect of diet on the preference.

This is a very interesting paper. The work seems to have been meticulously planned, executed and written up. The Supplementary Data provides a wealth of useful information on bar-coding, and the reference list of over 100 references (including one very comprehensive review at Ref 11) should also be very useful.

I have only relatively minor, editorial comments.

L42. P=0.104. State not statistically significant as L355

Fig. 2.  Remove dimensions as these are in text and the large arrows are confusing.  More helpful to point out wind direction, LED dome, landing platform and what all the other bits of equipment are.

Fig. 3.  Again dimensions are unnecessary. I can’t see the divider – is it obscured by the dimension arrow?

Table 1. Seems to be in twice

L255.  This is a bit ambiguous as written.  Presumably

Only the colours blue (477 nm) or yellow (588 nm) (Supplemental Figure S2) were presented to the thrips as single treatments, with four replicates per treatment.

L266. Side-by-side not really necesssary

L270. Delete combinations (just treatments). This was actually very confusing at first. Perhaps also change L267 three sets of derived colonies, to “six derived colonies”

L294. Evidence

L364. Is it worth pointing out here that for NLb there is no preference for blue or yellow whereas for DEb there seems to be a clear preference for blue in this setup, which is a little different, at least quantitatively, from the results on Experiment (i)?  This is ignored in the assessment of all three NL treatments “regardless of rearing conditions”.

L427. The above comment should also be acknowledged here because in no-choice  Experiment (i) NLb showed a significant preference for yellow over blue, but in choice Experiment (iii) NLb showed no preference. On the other hand, DEb showed a preference for blue in both experiments, not significant in Experiment (ii) but probably significant (?) in Experiment (iii).

L554. It is good that the authors have acknowledged that the percentage of thrips responding to the test stimuli is relatively small. The UV-A stimulus from the ceiling is also very much larger than the blue and yellow of the lamp. Could it also be that the thrips tend to fly upwards? And just to “light” in general, rather than a specific response to UV-A?

Author Response

Dear Reviewer 1:

We thank you for the positive and helpful reviews. We have tried to address all comments and concerns; all responses and modifications made to the revised version of manuscript have been made to the best of our abilities. You will find below our responses (in red) to each of your comments in the attached file. We have also made additional minor changes to the revised document to improve grammar, style, and clarity.

Reviewer 2 Report

This manuscript reports on studies to test whether previous conflicting research results on western flower thrips response to different colored traps (yellow vs blue) was due to methodology differences or potential differences in populations.  This is the first study to evaluate population differences. Using the same methodology, this study demonstrated differences in population response – lab colony from The Netherlands versus Germany – and also an associative learning response.  While the study was well designed and contributes new information to the literature, there are certain areas that need clarification and revision before publication.

11. Line 144 and throughout:  The different colonies are referred to as “country of origin.” This implies that country of origin may be a contributing factor to differential responses, which is not correct. Differences are likely due to inherent differences in strains or rearing conditions, not the country in which it originated. It is suggested this terminology be eliminated.

22.     New Fig. of reflectance: In the introduction or method section it may be valuable to include a reflectance wave length figure that includes not only the wave lengths/colors examined in this study, but also where green beans and chrysanthemum flower reflectance falls on the spectrum, and highlighting the region visible to humans.

33.     Lines 109 to 132: These two paragraphs could be combined and the text shortened. The selection process for selecting new strains was the same for each colony, plus Fig. 1 explains thing quite well.  Also, was there any reason for rearing NLc for 52 wks on chrysanthemum and DEc for only 6 wks?  This should at least be acknowledged in the text.

44.    Fig. 2 and wind tunnel description: A higher quality picture should be used. Also it is described as having 3 compartments, what was the purpose of the outside two compartments? In which compartment were thrips released?

55.    Experiment i: Why was only NLc used for exp. i? It seems logical that the DEc should also be tested.  Also, Table 1 is shown twice.

66.     Line 242: Please explain a Latinized resolvable block design, and what is mean by a block size of 6 days. This sentence is confusing – “…treatment was not always run at a particular point amongst replicates.”  Not sure what this means.

77.    Reference to colonies: It would be better to stick to NL and DE rather than “Dutch” and “German.” This would prevent the reader from having to go back and see which is which. My mind associated the D in DE for Dutch, which is not correct.

88.     Fig. 4. It would be helpful to use a dotted black line for the UV wavelength since it is invisible to us. Also, using a pink line could be construed as meaning this is pink in color, when in fact it’s not visible.

99.     Line 350: It is not accurate to say blue was preferred to yellow when differences were not significant.

110.  Line 362: What were the two main effects?

111.  Line 398: It seems that the UV-A attraction could have been eliminated by turning off the ceiling lights – it seems this should have been done.

12. Discussion:  The Discussion could be condensed considerably.  The first half of paragraph one is a repeat of the results and is not necessary. Also the 3rd paragraph goes into too much into population genetics of WFT that is only tangentially related to the main them of the manuscript.  In general, most pargraphs are very long and can be condensed without sacrificing discussion of the results.

13. Discussion:  The Discussion could be condensed considerably.  The first half of paragraph one is a repeat of the results and is not necessary. Also the 3rd paragraph goes into too much into population genetics of WFT that is only tangentially related to the main them of the manuscript.  In general, most pargraphs are very long and can be condensed without sacrificing discussion of the results.

Author Response

Dear Reviewer 2:

We thank you for the positive and helpful reviews. We have tried to address all your comments and concerns; all responses and modifications made to the revised version of manuscript have been made to the best of our abilities. You will find below our responses (in red) to each of your comments. We have also made additional minor changes to the revised document to improve grammar, style, and clarity.

Reviewer 3 Report

I have gone through the manuscript entitled, “Colour response in western flower thrips varies intraspecifically”. The manuscript describes behavioral responses of two thrips populations from Netherlands and Germany to blue and yellow colors.  The manuscript provides experimental evidence that thrips responses to colors are not universal and could vary from population to population. Therefore, trap color selection for thrips monitoring should be based on local population responses. In my opinion, the manuscript is well written and supports the conclusions presented here. I have minor edits and suggestions.

Comments and suggestions are in the attached file. 

Author Response

Dear Reviewer 3:

We thank you for the constructive comments. We have tried to address all comments and concerns. Suggested changes and modifications have been made to the best of our abilities and you will find them in the revised version of the manuscript. We have also made additional minor changes to the revised document to improve grammar, style, and clarity.

Round 2

Reviewer 2 Report

The authors have done a nice job of addressing reviewer comments and I recommend it be accepted.